# Dietary sugar inhibits satiation by decreasing the central processing of sweet taste

**Christina E May[1,2], Julia Rosander[3], Jennifer Gottfried[3], Evan Dennis[3], Monica Dus[1,2,3]\***

[1]The Neuroscience Graduate Program, The University of Michigan, Ann Arbor, United States; [2]Department of Molecular, Cellular and Developmental Biology, College of Literature, Science, and the Arts, The University of Michigan, Ann Arbor, United States; [3]The Undergraduate Program in Neuroscience, College of Literature, Science, and the Arts, The University of Michigan, Ann Arbor, United States

**Abstract** From humans to vinegar flies, exposure to diets rich in sugar and fat lowers taste sensation, changes food choices, and promotes feeding. However, how these peripheral alterations influence eating is unknown. Here we used the genetically tractable organism *D. melanogaster* to define the neural mechanisms through which this occurs. We characterized a population of protocerebral anterior medial dopaminergic neurons (PAM DANs) that innervates the β′2 compartment of the mushroom body and responds to sweet taste. In animals fed a high sugar diet, the response of PAM-β′2 to sweet stimuli was reduced and delayed, and sensitive to the strength of the signal transmission out of the sensory neurons. We found that PAM-β′2 DANs activity controls feeding rate and satiation: closed-loop optogenetic activation of β′2 DANs restored normal eating in animals fed high sucrose. These data argue that diet-dependent alterations in taste weaken satiation by impairing the central processing of sensory signals.

**\*For correspondence:**
mdus@umich.edu

**Competing interests:** The authors declare that no competing interests exist.

## Introduction

Consumption of diets high in sugar and fat decreases the perception of taste stimuli, influencing food preference and promoting food intake (*Bartoshuk et al., 2006*; *Sartor et al., 2011*; *Ahart et al., 2019*; *May et al., 2019*; *Weiss et al., 2019*; *Kaufman et al., 2018*). Recent studies have examined the effects of these diets on the sensitivity of the peripheral taste system and the intensity of taste experience (*May et al., 2019*; *Maliphol et al., 2013*; *Kaufman et al., 2018*; *Weiss et al., 2019*), but how exactly taste deficits increase feeding behavior is not known. Orosensory signals determine the palatability or 'liking' for foods (*Berridge and Kringelbach, 2015*), but they also promote meal termination via a process called 'sensory-enhanced (or mediated) satiety' (*Chambers et al., 2015*). Indeed, foods that provide longer and more intense sensory exposure are more satiating, reducing hunger and subsequent test-meal intake in humans (*Yeomans, 2017*; *Bolhuis et al., 2011*; *Ramaekers et al., 2014*; *Viskaal-van Dongen et al., 2011*; *Cecil et al., 1998*; *Forde et al., 2013*). Specifically, sensory signals are thought to function early in the satiety cascade (*Blundell et al., 1987*) by promoting satiation and bringing the on-going eating episode to an end (*Blundell et al., 2010*; *Bellisle and Blundell, 2013*). This is in contrast to nutrient-derived signals, which develop more slowly and consolidate satiety by inhibiting further eating after the end of a meal (*Blundell et al., 2010*; *Bellisle and Blundell, 2013*). We reasoned that if orosensory attributes like taste intensity are important to curtail a feeding event, then diet-dependent changes in taste sensation could promote feeding by impairing sensory-enhanced satiation. Here we investigated the relationship between diet composition – specifically high dietary sugar – the central processing of

**eLife digest** Obesity is a major health problem affecting over 650 million adults worldwide. It is typically caused by overeating high-energy foods, which often contain a lot of sugar. Consuming sugary foods triggers the production of a reward signal called dopamine in the brains of insects and mammals, which reinforces sugar-consuming behavior. The brain balances this with a process called 'sensory-enhanced satiety', which makes foods that provide a stronger sensation of sweetness better at reducing hunger and further eating.

High-energy food was scarce for most of human evolution, but over the past century sugar has become readily available in our diet leading to an increase in obesity. Last year, a study in fruit flies reported that a sugary diet reduces the sensitivity to sweet flavors, which leads to overeating and weight gain. It appears that this sensitivity is linked to the effectiveness of sensory-enhanced satiety. However, the mechanism linking diets high in sugar and overeating is still poorly understood. One hypothesis is that fruit flies estimate the energy content of food based on the degree of dopamine released in response to the sugar.

May et al. compared the responses of neurons in fruit flies fed a normal diet to those in flies fed a diet high in sugar. As expected, both groups activated the neurons involved in the dopamine reward response when they tasted sugar. However, when the flies were on a sugar-heavy diet, these neurons were less active. This was because the neurons responsible for tasting sweetness were activated less in flies fed a high-sugar diet, leading to a lowered response by the neurons that produce dopamine. The flies in these experiments were genetically engineered so that the dopamine-producing neurons could be artificially activated in response to light, a technique called optogenetics. When May et al. applied this technique to the flies on a sugar-heavy diet, they were able to stop these flies from overeating.

These findings provide further evidence to support the idea that a sugary diet reduces the brain's sensitivity to overeating. Given the significant healthcare cost of obesity to society, this improved understanding could help public health initiatives focusing on manufacturing food that is lower in sugar.

sweet taste signals, and satiation by exploiting the simple taste system and the conserved neuro-chemistry of the fruit fly *D. melanogaster*.

Like humans and rodents, *D. melanogaster* flies exposed to palatable diets rich in sugar or fat overconsume, gain weight, and become at-risk for obesity and develop phenotypes associated with metabolic syndrome (*Musselman and Kühnlein, 2018*). We recently showed that, in addition to promoting feeding by increasing meal size, consumption of high dietary sugar decreased the electro-physiological and calcium responses of the *Gr64f+* sweet sensing neurons to sweet stimuli, independently of weight gain (*May et al., 2019*). These physiological changes in the *Gr64f+* cells reduced the fruit flies' taste sensitivity and response intensity. Opto- and neurogenetics manipulations to correct the responses of the *Gr64f+* neurons to sugar prevented animals exposed to high dietary sugar from overfeeding and restored normal meal size (*May et al., 2019*). Thus, the diet-dependent dulling in sweet taste causes higher feeding in flies, but how does this happen? How do alterations in the peripheral sensory neurons modulate a behavior as complex as feeding? To better understand how this occurs, we decided to examine the effects of high dietary sugar and taste changes in the central processing of sweet stimuli by dopaminergic neurons (DANs). Indeed, while the neural pathways that bring sensory information from the periphery to higher order brain regions are unique across organisms, dopaminergic circuits process sweet taste information in humans, rodents, and fruit flies. Interestingly, the reinforcing effects of sugar taste and nutrient properties are relayed via distinct dopaminergic pathways in these organisms (*Yamagata et al., 2015*; *Huetteroth et al., 2015*; *Tellez et al., 2016*; *Thanarajah et al., 2019*). In flies, DANs in the Proto-cerebral Anterior Medial (PAM) cluster respond to the sweet sensory properties to signal sugar reward (*Burke et al., 2012*; *Liu et al., 2012*), reinforce short term appetitive memories (*Yamagata et al., 2015*; *Huetteroth et al., 2015*), and promote foraging and intake (*Tsao et al., 2018*; *Musso et al., 2019*). We hypothesized that diet-dependent impairments in the peripheral

responses to sugar could influence the way sweet taste information is transduced through PAM-DANs to affect feeding behavior and obesity risk.

Here we show that in flies fed a high sugar diet the presynaptic responses of a specific subset of PAM DANs to sweet taste are decreased and delayed. These changes are specific to sweet stimuli and mediated by high dietary sugar. Further, we report that the reduction in the central processing of sweet taste information increases the duration and size of meals: closed-loop optogenetic stimulation of a specific set of PAM DANs corrected meal size, duration, and feeding rate. Together, our results argue that diet-dependent alterations in the central processing of sweet sensory responses delay meal termination by impairing the process of sensory-enhanced satiation.

## Results

### Consumption of a high sugar diet decreases and delays the central processing of the sweet taste signal

We previously showed that the calcium responses of the sweet sensory neurons to sucrose were decreased in animals fed high dietary sugar (*May et al., 2019*; *Vaziri et al., 2020*). To ask if the transmission of the sweet taste signal out of these neurons was also lower, we expressed the genetically encoded vesicular release sensor *synaptobrevin-pHluorin* (*syb-pHluorin*) (*Poskanzer et al., 2003*) in the sweet taste neurons using the Gustatory Receptor 64f (*Gr64f*) GAL4 driver and measured the in vivo fluorescence from the presynaptic terminals in the Sub Esophageal Zone (SEZ) in response to stimulation of the proboscis with 30% sucrose (*Figure 1A*). We found that the *syb-pHluorin* fluorescent changes upon sugar presentation were markedly decreased when flies were fed a high sugar diet (SD, 30% sucrose) for 7 days, compared to age-matched flies fed a control diet (CD,~8% sucrose) (*Figure 1*). These data suggest that both the responses of the sweet sensing *Gr64f+* neurons to sugar and the transmission of the sweet taste signal are decreased by exposure to the SD.

Since the involvement of DANs in feeding behavior and in central processing of sensory information is a homologous feature across organisms (*Yamagata et al., 2015*; *Huetteroth et al., 2015*; *Tellez et al., 2016*; *Thanarajah et al., 2019*), we decided to center on this DAN circuit as a possible link between diet-dependent changes in sweet responses, higher feeding, and weight gain. In flies, DANs in the Protocerebral Anterior Medial (PAM) cluster that are labeled by the *R48B04-GAL4* transgene and innervate the β'2 and γ4 compartments of the Mushroom Body (MB), respond to sweet sensory properties (*Huetteroth et al., 2015*; *Yamagata et al., 2015*); neurons of this population also centrally reinforce water taste (*Lin et al., 2014*). Here we focused on the β'2 compartment

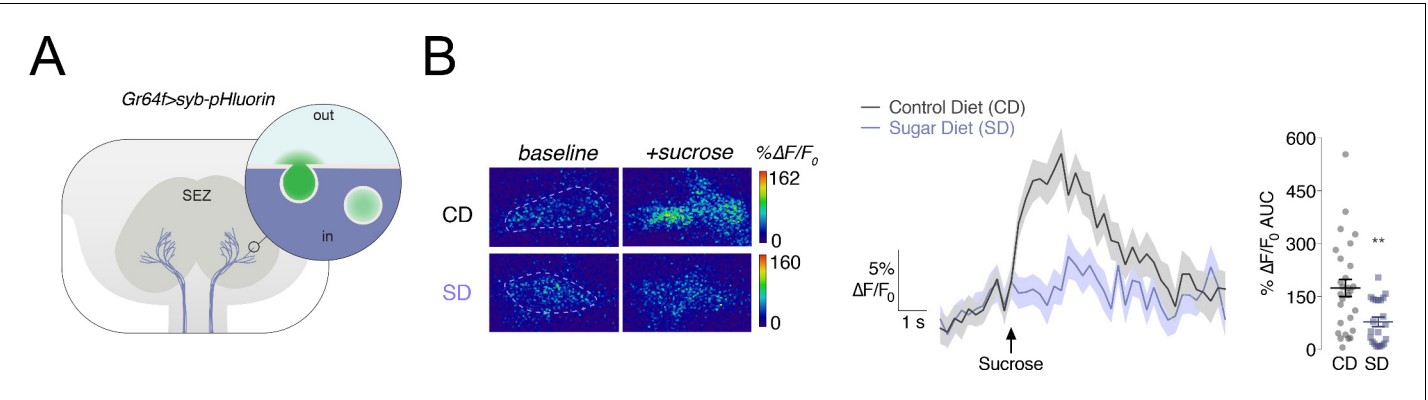

**Figure 1.** Vesicular release from the *Gr64f+* taste neurons in response to a sucrose stimulus is decreased in flies fed a high sugar diet. (**A**) Schematic of the subesophageal zone (SEZ), highlighting the *Gr64f+* neuron terminals in *lavender*. *Popout bubble* demonstrates increased fluorescence upon vesicular release. (**B**) *Left*, Representative frames just prior ('*baseline*') and during ('*+sucrose*') 30% sucrose stimulation from flies fed a control (CD) or sugar diet (SD). ROIs marked with dashed lines. *Center*, Mean %$\Delta$F/$F_0$ response traces, and *Right*, Area-under-the-curve (AUC) value of %$\Delta$F/$F_0$ responses when *Gr64f > syb* pHluorin flies fed a CD (*grey*) or SD (*lavender*) were stimulated with 30% sucrose on the labellum. n = 22–28; shading and error bars depict the standard error of the mean. Mann-Whitney test; **p<0.001.

because of its role in processing of the taste properties alone, compared to γ4 which is modulated by both taste and additional factors, such as internal state (*Lin et al., 2014*; *Yamagata et al., 2015*). In addition to labeling ~60 DANs in each PAM cluster, *R48B04* is expressed in other neurons, including the Pars Intercelebralis. To avoid or minimize potential confounding effects of its expression in other compartments, we used FlyLight to visually identify *GAL4* lines that label subsets of PAM-β'2, but have limited expression in other compartments (*Aso and Rubin, 2016*). We selected the split-*GAL4* line *MB301B* which had been implicated in foraging and feeding (*Tsao et al., 2018*; *Musso et al., 2019*) and which labels ~12 TH+ PAM-β2β'2a and only shows a few, sparse projections in the ventral nerve cord and SEZ (*Figure 2A* and *Figure 2—figure supplement 1A*). We used the presynaptically targeted *GCaMP6s::Bruchpilot::mCherry* (*Kiragasi et al., 2017*) to record the

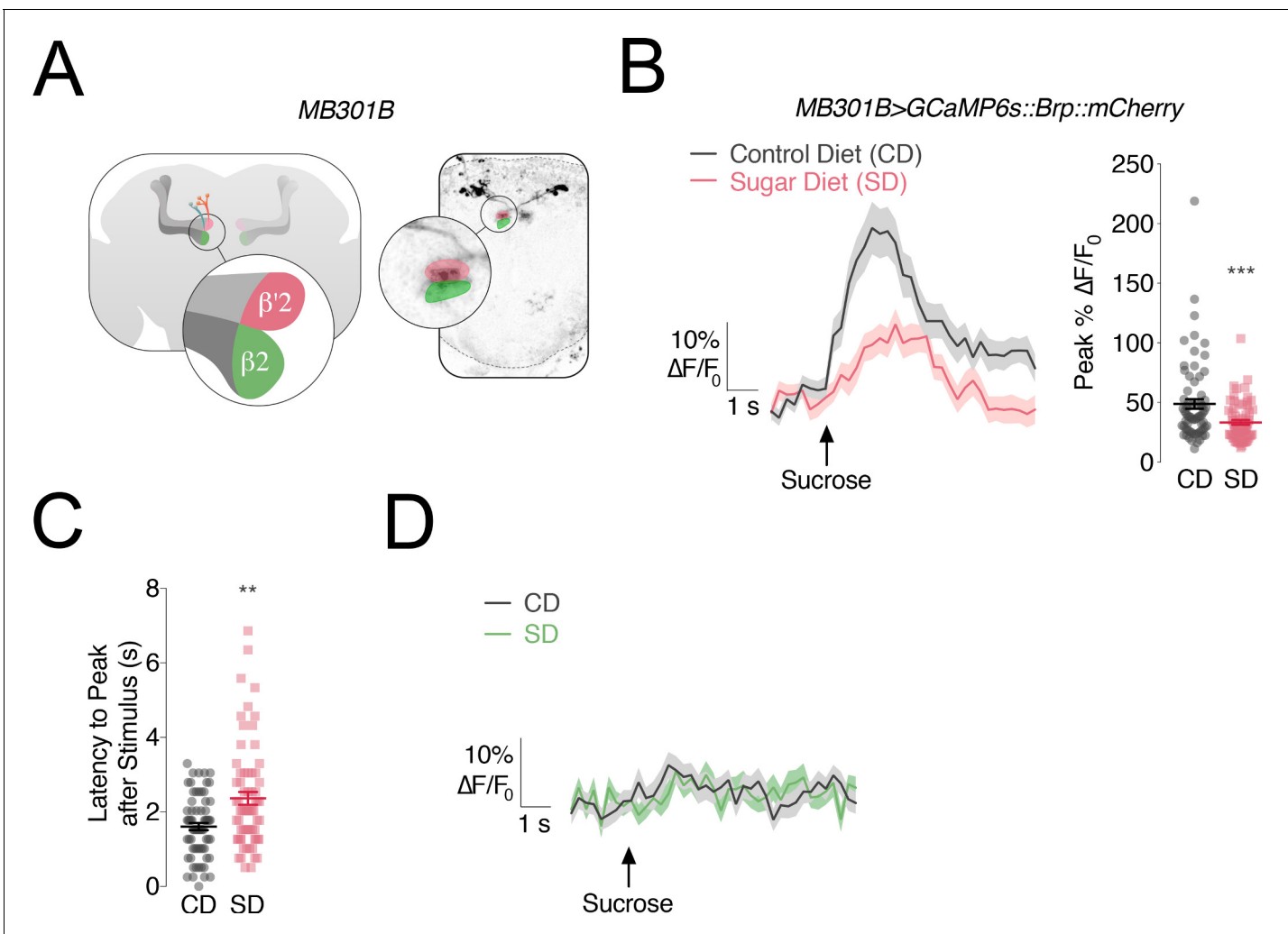

**Figure 2.** The responses of PAM-β'2 neurons to sweet stimuli change in flies fed a high sugar diet. (A) *Left,* Anatomy of the Mushroom Body (MB) of the *Drosophila melanogaster* brain, with α/β, α'/β' lobes in *greys,* and *MB301B* neurons in *rose* and *green; popout bubble,* schematic showing the β'2 (*rose*) and β2 (*green*) compartments in their respective MB lobes. *Right,* confocal image of *MB301B* neurons with β'2 (*rose*) and β2 (*green*) compartment expression. (B) *Left,* Mean %ΔF/F$_0$ traces, and *Right,* quantification of the maximum peak %ΔF/F$_0$ responses to 30% sucrose stimulation of the labellum in the β'2 compartment of *MB301B > GCaMP6s::Brp::mCherry* flies fed a control (CD, *grey*) and sugar diet (SD, *rose*). Shading and error bars are standard error of the mean. n = 67–70; Mann-Whitney test; ***p<0.001. (C) The delay in the calcium responses quantified as latency in seconds (s) to maximum peak ΔF/F$_0$ from the animals in B. n = 67–70; Mann-Whitney test; **p<0.01. (D) Mean %ΔF/F$_0$ traces for the responses to 30% sucrose stimulation of the labellum in the β2 compartment of *MB301B > GCaMP6s::Brp::mCherry* flies fed a control (CD, *grey*) and sugar diet (SD, *green*). n = 67–70; shading is standard error of the mean.

The online version of this article includes the following figure supplement(s) for figure 2:

**Figure supplement 1.** *MB301B* neurons are dopaminergic and respond to different concentrations of sucrose presentation to the labellum.

response of *MB301B* neurons to stimulation of the labellum with 30% sucrose. We observed an increase in signal in the β'2compartment (rose), showing that these PAM-β'2 neurons process sweet sensory information (*Figure 2B*, grey lines; *Figure 2—figure supplement 1B*). Next we measured the responses of *MB301B* neurons to sucrose taste in flies fed a SD for 7 days and we found a nearly 50% decrease (*Figure 2B*, rose lines; *Figure 2—figure supplement 1B*). Furthermore, when we looked at both the average and individual traces, we saw a ~ 600 millisecond delay in the peak responses to the sucrose stimulus delivery to the labellum (*Figure 2C*). A decrease in calcium responses also occurred when the proboscis was stimulated with a lower concentration of sucrose (5%), but we did not find a delayed response, suggesting that the changes in the timing of the processing may be unique to higher sugar concentrations (*Figure 2—figure supplement 1C and D*). No sugar taste responses were recorded in β2 (green), consistent with the idea that it is not involved in taste processing (*Figure 2D*; *Yamagata et al., 2015*). Thus, the central processing of sweet stimuli in PAM-β'2 *MB301B* neurons is both decreased and delayed by exposure to a high sugar diet.

## Alterations in PAM-β'2 responses are specific to high dietary sugar and sweet stimuli

The reduction and delay in central responses to sugar taste in PAM-β'2 DANs on a SD could be due either to the lower transmission of the sensory signal out of the peripheral sweet taste neurons (*Figure 1*) or to the metabolic side effects of the high nutrient diet. To differentiate between these possibilities, we took multiple approaches. In addition to sweet stimuli, PAM-β'2 neurons also respond to water (*Lin et al., 2014*); we reasoned that if high dietary sugar unspecifically changed the activity of the PAM-β'2, we would expect flies on the SD to also exhibit impaired central responses to water. However, the magnitude and timing of the β'2 response to water stimulation of the labellum were unchanged between flies on a CD or SD (*Figure 3A,B*, and *Figure 3—figure supplement 1A*; water stimulation was delivered in the same flies as in *Figure 2*). Thus, the decrease in PAM-β'2 responses in flies fed a SD is specific to the sweet sensory stimulus. This argues that the overall ability of these DANs to respond to stimuli is not generally affected, and the reduction observed on a SD could occur because of the diet-dependent changes in the sweet taste neurons in the periphery (*May et al., 2019* and *Figure 1*).

To further probe this question, we fed flies a high fat diet (FD), which has the same caloric content of the high sugar diet (SD) and promotes fat accumulation, but does not decrease the responses of the *Gr64f+* sensory neurons to sugar stimuli (*May et al., 2019*). If changes in PAM-β'2 responses to sugar taste occur because of the metabolic side-effects of high nutrient density (i.e, fat accumulation) – rather than via changes in the sweet sensory neurons' output – we would expect a FD to also induce PAM-β'2 dysfunction. However, a FD diet had no effect on the PAM-β'2 responses to sucrose or water stimulation of the labellum in *MB301B > GCaMP6s::Bruchpilot::mCherry* flies (*Figure 3C,D* and *Figure 3—figure supplement 1B,C*). Together, these two lines of evidence argue that the dysfunction in the processing of sweet taste stimuli in the PAM-β'2 neurons of flies on a SD is linked to alterations in the peripheral sensory processing of sugar taste caused by high dietary sugar.

To test this hypothesis more directly, we examined the effect of correcting sweet taste sensation on the responses of the PAM-β'2 *MB301B* neurons to sugar. To rescue the sweet taste deficits caused by a high sugar diet we fed flies an inhibitor of the metabolic-signalling enzyme O-GlcNAc-Transferase (OGT), which we previously found to be responsible for decreasing sweet taste on a SD (*May et al., 2019*). In accordance with our previous findings on OGT (*May et al., 2019*), supplementing the flies' diet with 75 µM of OSMI-1 (OGT-small molecule inhibitor 1) resulted in no changes in PER between a CD and SD (*Figure 3—figure supplement 1D*). In these flies, the calcium responses of PAM-β'2 neurons to sucrose stimulation of the labellum were identical in SD+OSMI and CD+OSMI flies. Although we cannot exclude the possibility that the OGT inhibitor also acted elsewhere outside the sensory neurons, our data support the idea that deficits in the peripheral responses drive impairments in the central processing of sweetness (*Figure 3E,F*). Together, these orthogonal lines of evidence suggest that the impairments in the central processing of sweet sensory information by DANs are mediated by deficits in peripheral sweet taste responses.

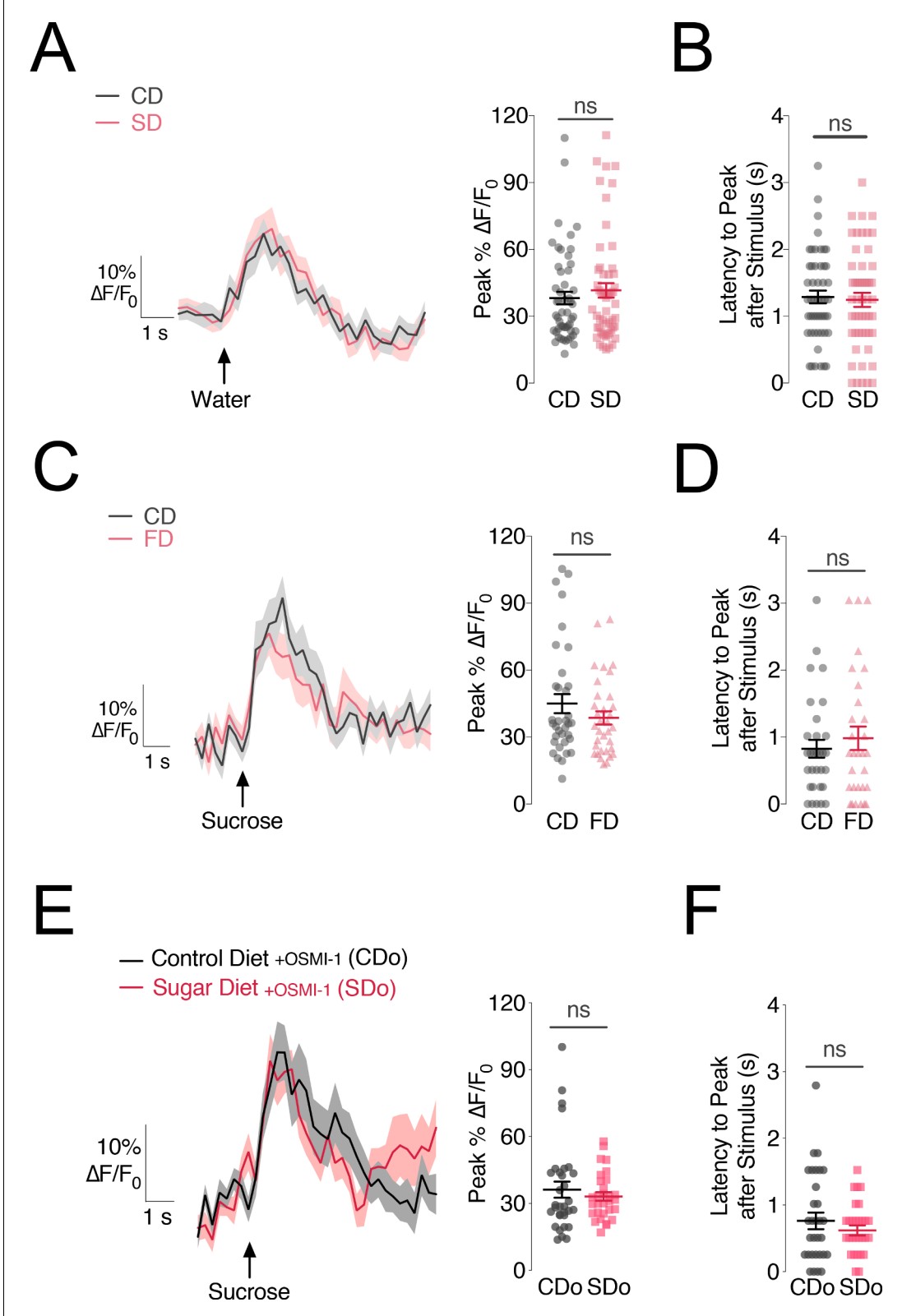

**Figure 3.** The changes in PAM-β'2 activity are specific to sugar stimuli and caused by deficits in sweet sensation. (A) *Left*, Mean %ΔF/F$_0$ traces and *Right*, quantification of the maximum peak %ΔF/F$_0$ responses to water stimulation of the labellum in the β'2 compartment of *MB301B > GCaMP6s::Brp:: mCherry* flies fed a control (CD, *grey*) and sugar diet (SD, *rose*), same animals as in **Figure 2**. n = 67–70; shading and error bars are standard error of the mean. Mann-Whitney test; no significance. (B) The delay in the calcium responses quantified as latency in seconds (s) to maximum peak ΔF/F$_0$

*Figure 3 continued on next page*

*Figure 3 continued*

response from the animals in A. n = 67–70; error bars are standard error of the mean. Mann-Whitney test; no significance. (C) *Left:* Mean $\%\Delta F/F_0$ response traces and *Right,* quantification of the maximum peak $\%\Delta F/F_0$ responses to 30% sucrose stimulation of the labellum in the β′2 compartment of *MB301B > GCaMP6s::Brp::mCherry* flies fed a control (CD, grey) or high fat diet (FD, rose) n = 31–32; shading and error bars are standard error of the mean. Mann-Whitney test; no significance. (D) Latency-to-peak response times for the animals in C. n = 31–32; error bars are standard error of the mean. Mann-Whitney test; no significance. (E) *Left,* Mean $\%\Delta F/F_0$ traces and *Right,* quantification of the maximum peak $\%\Delta F/F_0$ responses to sucrose stimulation of the labellum in the β′2 compartment of *MB301B > GCaMP6s::Brp::mCherry* flies fed a control (CD, *charcoal*) and sugar diet (SD, *red*) supplemented with 75 µM OSMI-1. n = 30–32; shading and error bars are standard error of the mean. Mann-Whitney test; no significance. (F) The delay in the calcium responses quantified as latency in seconds (s) to maximum peak $\Delta F/F_0$ response from the animals in E. n = 30–32; error bars are standard error of the mean. Mann-Whitney test; no significance.

The online version of this article includes the following figure supplement(s) for figure 3:

**Figure supplement 1.** Water responses are unchanged in animals fed a high fat diet.

**Figure supplement 2.** Knockdown of OGT in *MB301B* neurons has no effect on fat accumulation.

## Correcting the activity of PAM DANs rescues feeding behavior

We previously showed that a diet-dependent dulling of sweet taste drives higher feeding behavior and weight gain by increasing the size and duration of meals (*May et al., 2019*). Since sweet taste deficits underlie the changes in PAM-β′2 activity, we reasoned that impairments in the central processing of orosensory signals may also play a role in promoting higher feeding in animals fed a high sugar diet. Specifically, if PAM-β′2 neurons were critical for integrating sweet taste information into feeding decisions, then normalizing their activity may also prevent increased eating and weight gain when flies are exposed to a SD. To test this possibility we expressed the light-activated cation channel *ReaChR* (*Inagaki et al., 2014*) in the *MB301B* neurons, and used the optoFLIC, a feeding frequency assay (*Ro et al., 2014*) modified for closed-loop optogenetic stimulation (*May et al., 2019*), to normalize the change in activity of PAM-β′2 neurons only when the flies were interacting with the food starting at day 3. *MB301B > ReaChR* flies that did not receive retinal supplementation (ATR, all-*trans*-retinal is required to form a functional light-sensitive opsin) exhibited the characteristic increase in feeding behavior on 20% sucrose (*Figure 4A*, rose line); however, *MB301B > ReaChR* +ATR animals, which were activated by light, had stable feeding for 10 days (*Figure 4A*, peach line). Control animals on 20% sucrose had more feeding interactions per meal and longer meal duration with more days on the SD (*Figure 4B and C*, rose lines), consistent with our previous data (*May et al., 2019*). In particular, we found that a SD induced a lengthening of the peak-to-end of the meal by ~4 hr, suggesting that the satiation process is delayed in these animals (*Figure 4D*, rose line). However, feeding-paired stimulation of PAM-β′2 neurons stabilized the size and duration of the meal, as well as the time to satiation, over the entire duration of the experiment (*Figure 4B,C and D*, peach lines). Interestingly, stimulation of the *Gr64f+* sweet taste neurons also corrected these two aspects of meal structure (*May et al., 2019*). Importantly, flies in which these PAM-β′2 DANs were activated still developed taste deficits on a SD (*Figure 4—figure supplement 1A*), arguing against the possibility that PAM-β′2 DANs stimulation prevents increased feeding by rescuing the taste changes in the *Gr64f+* neurons. Instead, our data suggest that PAM-β′2 DANs modulate meal structure and feeding behavior by integrating the sensory signal from the periphery. Interestingly, identical stimulation of PAM-β′2 DANs in flies fed 5% sucrose resulted in higher feeding (*Figure 4— figure supplement 1B*), as previously showed with both sucrose food and water (*Musso et al., 2019*); this indicates the the context of animal's diet and the basal activity state of PAM-β′2 DANs are important to control eating.

In accordance with the stable feeding patterns recorded on the optoFLIC in animals fed a SD, we found that activation of PAM-β′2 DANs also prevented diet-induced obesity in animals fed high dietary sugar (*Figure 4—figure supplement 1C*). Interestingly, PAM-β′2 DANs labeled by *MB301B* seem to play a unique role in this process. Activation of different subpopulations of PAM-β′2 with eight distinct *GAL4* transgenes (*Aso and Rubin, 2016*; *Aso et al., 2014b*) (*MB056B, MB109B, MB042B, MB032B, MB312B, MB196B, MB316B*, some of these also express in γ4) failed to rescue diet-induced obesity (*Figure 4—figure supplements 1D* and *2A*). Comparison of the anatomy of these lines showed that *MB301B* projects more anteriorly and ventrally than these other lines (*Figure 4—figure supplement 2A*). However, the ventral expression mostly regards the β2 compartment, which it is shared with only one of the other lines (*MB032B*); meanwhile the *MB301B* β′2

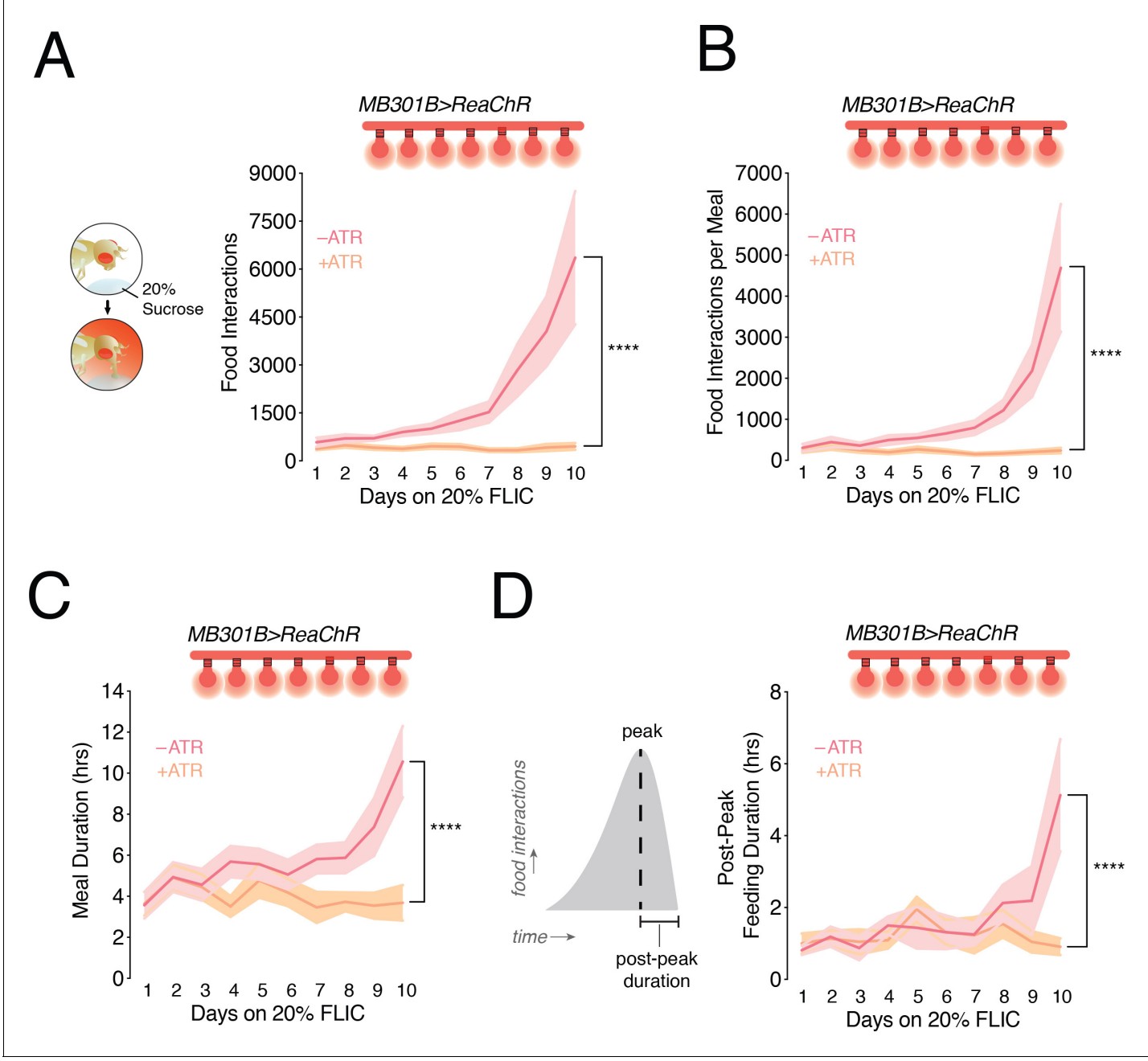

**Figure 4.** Closed-loop optogenetic activation of PAM-β'2 neurons corrects meal size and duration in flies fed a high sugar diet. (**A**) *Left:* Conceptual schematic for the closed-loop optogenetic FLIC (optoFLIC), wherein a fly feeding on the 20% sucrose food triggers delivery of the red light during the food interaction. *Right:* Mean number of food interactions per day for *MB301B > ReaChR* flies fed 20% sucrose on the optoFLIC. Closed-loop light delivery was started on day 3 (indicated with *red light bulbs*). Control flies were not fed retinal (-ATR, *rose*), while experimental animals were fed retinal food before starting the experiment on the optoFLIC (+ATR, *peach*). n = 8–11; shading is standard error of the mean. Two-way Repeated Measure (RM) ANOVA; ****p<0.0001, Time by Retinal-treatment interaction. (**B**) The size of the evening meal measured as the number of food interactions per meal for animals in A. n = 8–11; shading is standard error of the mean. Two-way RM ANOVA; ****p<0.0001, Time by Retinal-treatment interaction. (**C**) The duration of the evening meal for animals in A. n = 8–11; shading is standard error of the mean. Two-way RM ANOVA; ****p<0.0001, Time by Retinal-treatment interaction. (**D**) *Left*, schematic of an evening meal, and *Right*, mean duration of the portion of the evening meal after the peak (satiation) in animals from A. n = 8–11; shading is standard error of the mean. Two-way RM ANOVA; ****p<0.0001, Time by Retinal-treatment interaction.

The online version of this article includes the following figure supplement(s) for figure 4:

**Figure supplement 1.** The effects of activating different PAM neurons on diet-induced obesity.
**Figure supplement 2.** Anatomical comparison of a subset of PAM neurons.

expression, while more anterior than other lines, does overlap with some (*MB032B*, *MB196B*, *MB042B*). (*Figure 4—figure supplement 2A*). Further, flies with activation of nutrient-responsive PAM DANs (*Yamagata et al., 2015*; *Huetteroth et al., 2015*), which express in β2, still accumulated fat as controls when fed high dietary sugar, suggesting that effects of *MB301B* neuron activation come from the sweet-responsive β'2 compartment (*Figure 4—figure supplement 1E*).

## PAM-β'2 activity modulates the feeding rate during a meal

Since the FLIC records feeding interactions every 200 milliseconds (*Ro et al., 2014*), we used this information to look at how feeding rate changed during a meal, as this has been linked to the process of satiation. To do this, we first calculated the number of feeding events per meal, where a feeding event is defined as a succession of consecutive feeding interactions above an established signal threshold, (see Materials and methods, and *Ro et al., 2014*). We next divided the number of events per meal by the duration of each meal per day to obtain a feeding rate and to control for the fact that meals last longer on a SD. We found that both the feeding events per meal and the feeding rate increased with chronic exposures to high dietary sugar (*Figure 5A and B*). However, optogenetic stimulation of PAM-β'2 prevented these increases and maintained a stable number of events and a constant feeding rate per meal over the duration of the experiment. We next examined whether the feeding rate changed during the course of the meal, by calculating it *before* and *after* the peak of meal feeding (*Figure 5C*, diagram). The feeding rate past the peak of the meal increased with time in animals fed 20% sucrose, but stayed the same in flies with activation of PAM-β'2 neurons (*Figure 5C*). Interestingly, the pre-peak eating also increased gradually with exposure to high dietary sugar (*Figure 5D*). Together, these data suggest that diet-dependent impairments in PAM-β'2 neurons promote overfeeding by impairing satiation, and specifically by affecting the feeding rate during a meal. Since, PAM-β'2 neurons process sensory experiences from the periphery, our experiments argue that this phenomenon is connected to sensory-enhanced satiation. Together we propose that the central processing of sensory experiences during a meal by PAM-β'2 DANs, controls feeding rate and sensory-enhanced satiation. This process is altered by high dietary sugar, leading to an attenuated satiation process and higher feeding (*Figure 5E*).

## Discussion

In this study we found that diet-dependent changes in sensory perception promote feeding and weight gain by impairing the central dopaminergic processing of sweet taste information. When animals consume a high sugar diet, the responses to sweet taste of a distinct population of PAM DANs innervating the β'2 compartment of the MB are decreased and delayed. These alterations in dopaminergic processing increase the eating rate and extend the duration of meals, leading to attenuated satiation, higher feeding, and weight gain (*Figure 5E*). Interestingly, we observed a reduction in PAM DAN responses only when flies ate diets that resulted in sweet taste deficits; consumption of an equal calorically-rich lard diet that did not impact taste had no effect on the PAM DANs responses. Similarly, animals fed high dietary sugar exhibited differences in PAM-β'2 responses to sweet, but not water taste stimuli, reinforcing the idea that PAM DAN alterations occur because of lower signal transmission from the sensory neurons (*Figure 5E*). Indeed, correcting sweet taste deficits by feeding fruit flies an inhibitor of the enzyme O-GlcNAc Transferase – which we previously found to be required for taste impairments– prevented impairments in PAM-β'2 responses, although we cannot exclude that this could also be due to its effect beyond the sensory neurons (but not in the *MB301B* neurons, *Figure 3—figure supplement 2A*). Here, we propose a model where diet-dependent changes in taste intensity and sensitivity reduce the central processing of sensory stimuli to cause weaker and attenuated satiation. Interestingly, our anatomical analysis of several PAM-β'2 DANs lines showed that the neurons labeled by *MB301B* minimally overlap with other split-*GAL4* lines purported to express in β'2 (A degree of overlap is not unexpected and does not necessarily imply that the neurons accessed by these driver lines perform identical functions.) *MB301B* is largely distinct from these other lines both ventrally and anteriorly. Ventral expression of *MB301B* enters the β2 compartment, which has been implicated in nutrient reward; however, this compartment is not responsive to sweetness and its activation still resulted in diet-induced obesity. Thus, MB301B expression in anterior β'2 could represent an unique MB compartment that is part of a circuit dedicated to sweet taste processing and feeding behavior.

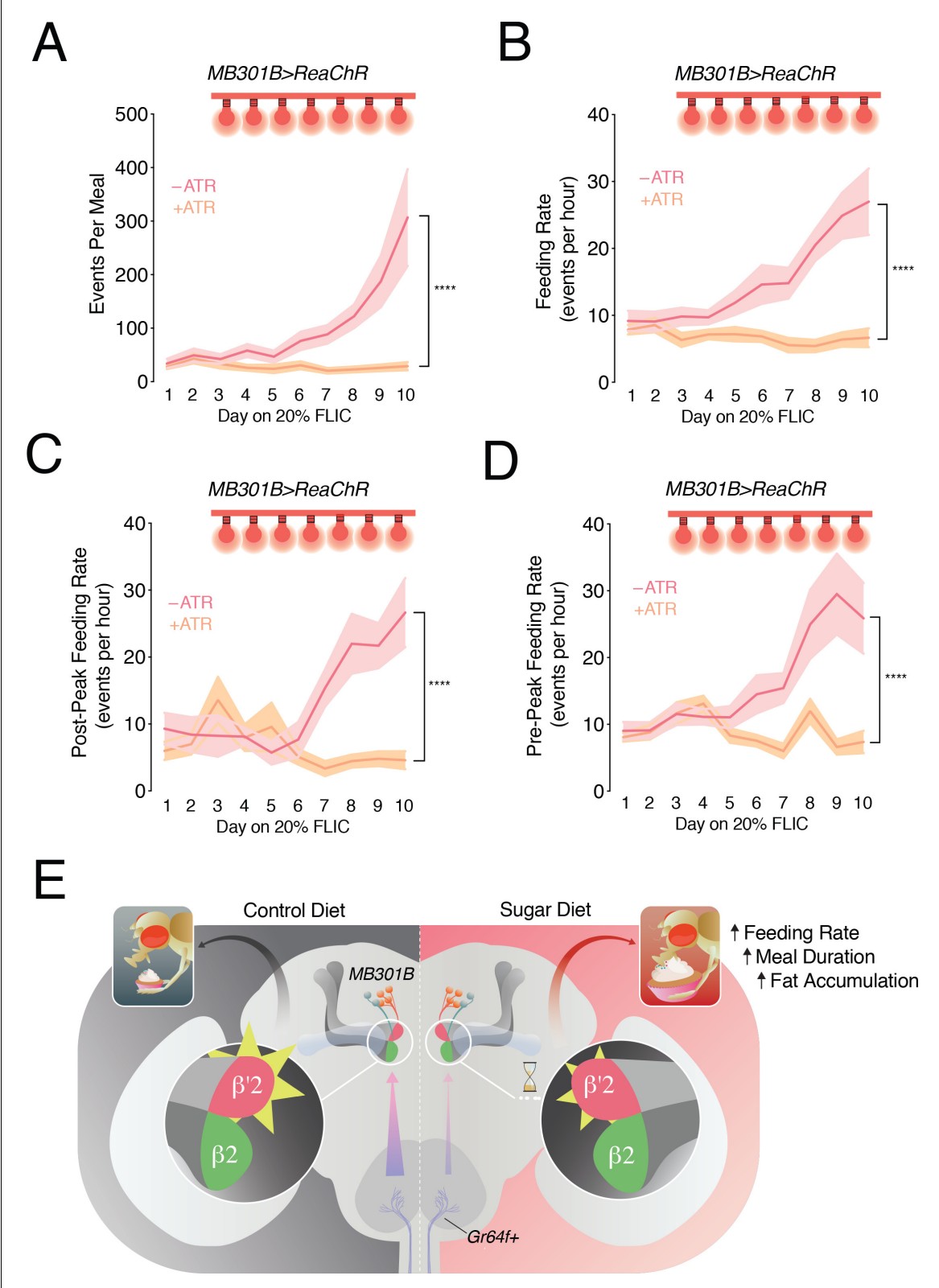

**Figure 5.** Feeding rate is modulated by a high sugar diet and controlled by the activity of PAM-β'2 neurons. (**A**) The mean of total feeding events per meal in *MB301B > ReaChR* flies with (- ATR, *rose*) or without (+ATR, *peach*) retinal pretreatment. A feeding event is calculated as the number of consecutive licks above and below the signal threshold (see Materials and methods). n = 8–11; shading is standard error of the mean. Two-way Repeated Measures (RM) ANOVA; ****p<0.0001, Time by Retinal-treatment interaction. (**B**) The feeding rate per meal, calculated as the mean number
*Figure 5 continued on next page*

*Figure 5 continued*

of events per hour of mealtime in the animals from A. n = 8–11; shading is standard error of the mean. Two-way RM ANOVA; ****p<0.0001, Time by Retinal-treatment interaction. (C) Quantification of the mean feeding rate *after* the peak of the meal in animals from A. n = 8–11; shading is standard error of the mean. Two-way RM ANOVA; ****p<0.0001, Time by Retinal-treatment interaction. (D) Quantification of the mean feeding rate *before and including* the peak of the meal from flies in A. n = 8–11; shading is standard error of the mean. Two-way RM ANOVA; ****p<0.0001, Time by Retinal-treatment interaction. (E) Model of the sweet taste and PAM DAN circuit changes when flies are fed a control (*left*) or high sugar diet (*right*): a decrease in the output of the *Gr64f+* neurons (*lavender axons, arrows*) contributes to a decrease (*yellow rays*) and a delay (*hourglass*) in the central processing of sweet taste information in the PAM-β'2 terminals (*rose*), promoting higher feeding.

A weakness of the current study is that we were unable to follow the transmission of the taste signal from the primary sensory neurons through the different circuits that eventually communicate with PAM. Studies that will identify taste projection neurons genetically will allow us to further probe this point in the future. Further, while the split-*GAL4* line used in this study expresses in PAM-β2β'2 neurons in the central brain, it also labels a few projections in the ventral nerve cord/SEZ, which could also contribute to some of the effects measured here. Interestingly, two studies previously found that neurogenetic or closed-loop optogenetic activation of *MB301B* PAM-β2β'2 neurons resulted in an increase in foraging behavior (*Tsao et al., 2018*) and higher feeding to both water and sucrose substrates (*Musso et al., 2019*), respectively. We also measured an increase in feeding behavior with closed-loop optogenetic activation of *MB301B* neurons in flies fed a control diet (5% sucrose), confirming these observations. However, applying the same optogenetic protocol when animals consumed a high sugar diet resulted in lower eating and a protection from diet-induced obesity. Since these neurons have lower activity in flies on a high sugar diet, we propose that the optogenetic stimulation in this context functions as a normalization of the activity, rather than activation in the absence of the stimulus. This suggests that variations in relative PAM DANs activity, rather than their absolute output, may modulate feeding behavior in flies exposed to high dietary sugar. It will also be interesting to ask how the activity of these neurons relates to other aspects of feeding behavior, such as the acceptance of low quality or bitter foods.

Studies in rodents and humans have delineated the importance of sensory signals to modulate satiation and terminate meals. This process, termed sensory-enhanced satiation (*Chambers et al., 2015*), plays an early role in the satiety cascade before post-oral nutrient-derived signals consolidate satiety (*Bellisle and Blundell, 2013*; *Blundell et al., 1987*). Data show that higher sensory intensity and oral exposure promote stronger satiation (*Bolhuis et al., 2011*; *Ramaekers et al., 2014*). For example, high sensory characteristics, such as saltiness and sweetness, enhanced the satiating effect of both low and high energy test drinks (*Yeomans and Chambers, 2011*; *Yeomans et al., 2014*), decreased consumption of pasta sauce (*Yeomans, 1998*; *Yeomans, 1996*), yoghurt (*Vickers et al., 2001*) and tea (*Vickers and Holton, 1998*). However, the neural basis for this phenomenon is unknown. Here we characterized the circuit-based mechanisms of sensory-enhanced satiation by exploiting the simplicity of the fruit fly system. We show that sensory-enhanced satiation involves the central dopaminergic processing of peripheral sweet taste stimuli by a dedicated group of PAM-β'2 neurons. Given the role of PAM DANs transmission in reinforcing appetitive memories (*Burke et al., 2012*; *Liu et al., 2012*), this discovery is significant because it suggests that satiation may involve a learning or rewarding component and that diet composition may direct food intake by influencing this aspect. Indeed, sensory cues function as a predictor of nutrient density and set expectations for how filling different types of foods should be (*Chambers et al., 2015*; *McCrickerd and Forde, 2016*; *Yeomans, 2017*). This information could be used to modulate the feeding rate during the meal and initiate the process of meal termination without relying uniquely on nutrient-derived cues, which arrive later (*Bellisle and Blundell, 2013*; *Blundell et al., 1987*). The idea that sensory cues could set cognitive expectations about the fullness of future meals is also in line with the known roles of DA in promoting the formation of appetitive memories. In fruit flies, PAM DANs promote the formation of short-term associative memories based on taste and long-term associative memories based on nutrient density by modulating plasticity of the postsynaptic Mushroom Body Output Neurons (MBONs) (*Cohn et al., 2015*; *Owald et al., 2015*). MBONs are, in turn, connected to pre-motor areas like the Central Complex (*Aso et al., 2014a*) – the fly genetic and functional analog of the basal ganglia (*Strausfeld and Hirth, 2013*) – providing an anatomical route to modulate aspects of feeding such as proboscis extension (*Chia and Scott, 2019*), the analogue of licking or chewing rate.

Interestingly, some MBONs receive input from both the taste (β'2) and nutrient (γ5) compartments, raising the possibility that sensory and nutrient memories may be integrated in the same cells to regulate different aspects of the satiety cascade (satiation vs. satiety). In flies, the mode and timing of DA delivery onto the MBONs is critical to establish the strength and valence of the associations (*Handler et al., 2019*). The delay and decrease we measured in animals on a high sugar diet could impair MBON synaptic plasticity and the formation of new appetitive memories (*Cohn et al., 2015*; *Owald et al., 2015*). If this is the case, we would expect that flies on this diet may be insensitive to new learning, use old food memories to predict the filling effects of the meal, and thus overshoot their food intake. This is consistent with the idea elegantly espoused by *Kroemer and Small, 2016* who explain the decrease in DA transmission with diet or obesity in a reinforcement learning framework. A different possibility, however, is that alterations in PAM DAN processing are not related to reinforcement learning per se, but instead to a decrease in overall reward receipt. In this light, sensory signals would cue reward not learning, and the pleasure experienced during eating would promote satiation and curb food intake. The idea that decreases in the sensitivity of the reward system increases food intake has been described as the 'reward deficit' theory of obesity (*Wang et al., 2002*), which also draws a parallel between the effects of drugs of abuse and that of sugar on the brain. Our results are consistent with both reinforcement learning and reward deficit scenarios, as well as with other integrated theories of obesity (*Stice and Yokum, 2016*); future experiments examining the role of circuits downstream of PAMs, and especially the involvement of MBONs, will differentiate between these possibilities. In addition to contributing to the current body of evidence connecting diet with DA alterations in mammals (*DiFeliceantonio and Small, 2019*; *Kroemer and Small, 2016*; *Geiger et al., 2009*; *Friend et al., 2017*; *van de Giessen et al., 2013*), our results also show that at least some of these alterations are due to diet, and not obesity. In particular, we speculate that some of the changes in DA transmission observed with diet exposure in rodents and humans may be due to impairments in sensory processing, since humans and rodents also process the taste and nutritive properties of sugar separately (*Tellez et al., 2016*; *Thanarajah et al., 2019*). It will be particularly interesting to test whether mimicking the effects of a high sugar diet on these DANs using optogenetics will promote feeding behavior.

In conclusion, our experiments demonstrate that by reducing peripheral taste sensation, a high sugar diet impairs the central DA processing of sensory signals and weakens satiation. These studies forge a causal link between sugar – a key component of processed foods – taste sensation, and weakened satiation, consistent with the fact that humans consume more calories when their diets consist of processed foods (*Hall et al., 2019*). Given the importance of sensory changes in initiating this cascade of circuit dysfunction, understanding how diet composition mechanistically affects taste is imperative to understand how the food environment directs feeding behavior and metabolic disease.

## Materials and methods

**Key resources table**

| Reagent type (species) or resource | Designation | Source or reference | Identifiers | Additional information |
|---|---|---|---|---|
| Genetic reagent (*D. melanogaster*) | Gr64f-GAL4 | H. Amrein; *Kwon et al., 2011* | RRID:BDSC_57669 | Flybase symbol: P{Gr64f-GAL4.9.7}5 |
| Genetic reagent (*D. melanogaster*) | UAS-n-Syb-pH (pHluorin) | B. Ye; *Poskanzer et al., 2003* | n/a | n/a |
| Genetic reagent (*D. melanogaster*) | MB301B-GAL4 | Bloomington *Drosophila* Stock Center; *Aso et al., 2014b* | RRID:BDSC_68311 | Flybase symbols: P{y[+t7.7] w[+mC]=R71D01-p65.AD}attP40; P{y[+t7.7] w[+mC]=R58 F02-GAL4.DBD}attP2 |
| Genetic reagent (*D. melanogaster*) | UAS-GCaMP6S:: Brp::mCherry | Bloomington *Drosophila* Stock Center; *Kiragasi et al., 2017* | RRID:BDSC_77131 | Flybase symbol: P{w[+mC]=UAS-GCaMP6s.brpS.mCherry}2 |

*Continued on next page*

*Continued*

| Reagent type (species) or resource | Designation | Source or reference | Identifiers | Additional information |
|---|---|---|---|---|
| Genetic reagent (*D. melanogaster*) | UAS-ReaChR | Bloomington *Drosophila* Stock Center; *Inagaki et al., 2014* | RRID:BDSC_53741 | Flybase symbol: P{y[+t7.7] w[+mC]=UAS-ReaChR}attP40 |
| Genetic reagent (*D. melanogaster*) | UAS-NaChBac | *Nitabach, 2006* | RRID:BDSC_9469 | Flybase symbol: P{UAS-NaChBac}2 |
| Genetic reagent (*D. melanogaster*) | MB032B-GAL4 | Bloomington *Drosophila* Stock Center; *Aso et al., 2014b* | RRID:BDSC_68302 | Flybase symbols: P{y[+t7.7] w[+mC]=R30 G08-p65.AD}attP40; P{y[+t7.7] w [+mC]=ple-GAL4.DBD}attP2 |
| Genetic reagent (*D. melanogaster*) | MB042B-GAL4 | Bloomington *Drosophila* Stock Center; *Aso et al., 2014b* | RRID:BDSC_68303 | Flybase symbols: P{y[+t7.7] w[+mC]=R58E02 -p65.AD}attP40/CyO; P{y[+t7.7] w[+mC]=R22E04-GAL4.DBD}attP2 |
| Genetic reagent (*D. melanogaster*) | MB056B-GAL4 | Bloomington *Drosophila* Stock Center; *Aso et al., 2014b* | RRID:BDSC_68276 | Flybase symbols: P{y[+t7.7] w[+mC]=R76 F05-p65.AD}attP40; P{y[+t7.7] w[+mC] =R80 G12-GAL4.DBD}attP2 |
| Genetic reagent (*D. melanogaster*) | MB109B-GAL4 | Bloomington *Drosophila* Stock Center; *Aso et al., 2014b* | RRID:BDSC_68261 | Flybase symbols: P{y[+t7.7] w[+mC ]=R76 F05-p65.AD}attP40; P{y[+t7.7] w[+mC]=R23 C12-GAL4.DBD}attP2 |
| Genetic reagent (*D. melanogaster*) | MB196B-GAL4 | Bloomington *Drosophila* Stock Center; *Aso et al., 2014b* | RRID:BDSC_68271 | Flybase symbols: P{y[+t7.7] w[+mC]=R58E02 -p65.AD}attP40/CyO; P{y[+t7.7] w[+mC]=R36B06-GAL4.DBD}attP2 |
| Genetic reagent (*D. melanogaster*) | MB312B-GAL4 | Bloomington *Drosophila* Stock Center; *Aso et al., 2014b* | RRID:BDSC_68314 | Flybase symbols: P{y[+t7.7] w[+mC]=R58E02 -p65.AD}attP40/CyO; P{y[+t7.7] w[+mC]=R10 G03-GAL4.DBD}attP2 |
| Genetic reagent (*D. melanogaster*) | MB316B-GAL4 | Bloomington *Drosophila* Stock Center; *Aso et al., 2014b* | RRID:BDSC_68317 | Flybase symbols: P{y[+t7.7] w[+mC]=R58E02-p65.AD}attP40/CyO; P{y[+t7.7] w[+mC]=R93 G08-GAL4.DBD}attP2 |
| Genetic reagent (*D. melanogaster*) | UAS-mCD8-RFP, LexAop-mCD8-GFP | Bloomington *Drosophila* Stock Center; *Pfeiffer et al., 2010* | RRID:BDSC_32229 | Flybase symbols: P{y[+t7.7] w[+mC]=10XUAS-IVS-mCD8::RFP}attP18; P{y[+t7.7] w[+mC]=13XLexAop2-mCD8::GFP}su(Hw)attP8 |
| Genetic reagent (*D. melanogaster*) | UAS-mCD8-GFP | A.-S. Chiang; *Dus et al., 2015* | n/a | |
| Genetic reagent (*D. melanogaster*) | w[1118]-CS | A. Simon | n/a | |
| Commercial assay or kit | Pierce BCA Protein Assay Kit | Thermo Scientific | Cat. #23225 | |
| Commercial assay or kit | Triglyceride LiquiColor Test (Enzymatic) | Stanbio | Ref. # 2100–430 | |
| Chemical compound, drug | D-sucrose | Fisher Scientific | BP220-10 | |
| Chemical compound, drug | all-*trans*-retinal | Sigma-Aldrich | R2500-100MG, CAS: 116-31-4 | |

*Continued on next page*

*Continued*

| Reagent type (species) or resource | Designation | Source or reference | Identifiers | Additional information |
|---|---|---|---|---|
| Software, algorithm | Olympus FluoView FV1200-ASW 4.2 | Olympus Life Science | RRID:SCR_014215 | |
| Software, algorithm | FLIC Monitor | FLIC support; *Ro et al., 2014* | RRID:SCR_018387 | |
| Software, algorithm | RStudio | RStudio, Inc | RRID:SCR_000432 | |
| Software, algorithm | FLIC analysis R code | FLIC support; *Ro et al., 2014*; *May et al., 2019* | RRID:SCR_018386 | |
| Software, algorithm | Fiji | ImageJ | RRID:SCR_002285 | |

## Fly lines and preparation

All flies were maintained at 25°C in a humidity-controlled incubator with a 12:12 hr light/dark cycle. For all experiments, males were collected under $CO_2$ anesthesia, 2–4 days following eclosion, and housed in groups of 20–30 within culture vials. The *GAL4/UAS* system was used for cell-type specific expression of transgenes. Stocks used are listed in the Key Resources Table. As control we used $w^{1118}$*Canton-S* flies (gift from Anne Simon, University of Western Ontario), which were obtained by backcrossing a $w^{1118}$ strain (Benzer lab, Caltech) to *Canton-S* (Benzer lab, Caltech) for 10 generations.

## Dietary manipulations

Flies were transferred to each diet 2–4 days after eclosion in groups of 30 animals per vial and fed on experimental diets (SD or FD) for 7 days with age-matched controls on CD.

The composition and caloric amount of each diet was as below:

- 'Control Diet/CD' was a standard cornmeal food (Bloomington Food B recipe), with approx. 0.6 cal/g.
- 'Sugar Diet/SD' was 30 g of table sugar added to 89 g Control Diet for 100 mL final volume of 30% sucrose w/v, with approx. 1.4 cal/g.
- 'Fat Diet/FD' was 10 mL of melted lard added to 90 mL of liquid Control Diet for 100 mL final volume of 10% lard v/v, with approx 1.4 cal/g.
- For diets supplemented with OSMI-1, the inhibitor was dissolved in 55% DMSO for a stock concentration of 500 µM, and then diluted 3:20 in liquid Control or Sugar Diet for a final concentration of 75 µM in food.
- For diets supplemented with all-*trans*-retinal, retinal was dissolved in 95% EtOH for a stock concentration of 20 mM, then diluted 1:100 in liquid Control Diet for a final concentration of 200 µM in food.
- Diets on the FLIC were 5% and 20% w/v D-sucrose in 4 mg/L $MgCl_2$. 20% was used instead of 30% to avoid potential problems with viscosity of the higher sucrose concentration food; 20% sucrose recapitulates the effects of 30% sucrose (*May et al., 2019*).

## In vivo imaging

Adult age-matched male flies, following 7 days of CD or SD, were fasted on a wet Kimwipe for 18–24 hr before prepping for in vivo confocal laser imaging. As previously described (*May et al., 2019*; *LeDue et al., 2015*), the preparation consisted of a fly affixed to a 3D-printed slide with melted wax around the head and on the dorsal part of the thorax. Distal tarsal segments were removed to prevent interference of the proboscis stimulus, and the proboscis was wax-fixed fully extended with the labellum functional and clear of wax so that proboscis contraction and extension could not perturb the brain's position. A glass coverslip was placed such that artificial hemolymph (108 mM NaCl, 8.2 mM $MgCl_2$, 4 mM $NaHCO_3$, 1 mM $NaH_2PO_4$, 2 mM $CaCl_2$, 5 mM KCl, 5 mM HEPES) placed over the head did not touch the proboscis. Data were acquired with a FV1200 Olympus confocal microscope, a 20x water immersion objective, and a rate of 0.254 s per frame. Stimuli consisted of a brief touch of a small Kimwipe soaked in milliQ water or 30% sucrose solution to the labellum. Responses to

both sucrose and water were measured in the same fly. For calcium imaging experiments, n counts each ROI, of which there are two per fly.

## Optogenetic stimulation for Fly-to-Liquid-food Interaction Counter (optoFLIC)

OptoFLIC was run as previously described (*May et al., 2019*). Briefly, adult flies 3–5 days past eclosion were placed on ATR food and kept in the dark for 3 days until starting the optoFLIC. optoFLIC experiments were run in an incubator with consistent 25°C and 30–40% humidity, on a dark/dark light cycle to prevent ambient-light activation of the ReaChR. Following two days recording of feeding activity on the FLIC food without LED activation, a protocol for closed-loop feeding-triggered LED activation was begun. The LED activation protocols were as follows:

For experiments with *MB301B > ReaChR*, 200 ms of red (~627 nm) light pulsing at frequency 60 Hz and with a pulse width of 4 ms was triggered by every food interaction signal over 10. n = 1 is a single animal.

## Immunofluorescence staining

Immunofluorescence protocol was performed as described in *Dus et al., 2015*. Briefly, brains were dissected in 1xPBS from male *MB301B > RFP* flies 3–5 days post-eclosion, then fixed in 4% paraformaldehyde in 1xPBS for 20 min, blocked in blocking buffer (10% normal goat serum, 2% Triton X-100 in 1xPBS), and incubated overnight at RT in anti-TH (rabbit polyclonal Ab from Novus Bio) 1:250 in dilution buffer (1% normal goat serum, 0.25 Triton X-100 in 1xPBS). Secondary antibody was goat anti-rabbit Alexa Fluor 488 diluted 1:500 in dilution buffer, and brains were washed then incubated with secondary antibody overnight at RT. Brains were mounted in FocusClear between two coverslips and imaged within 24 hr.

## Triacylglyceride (TAG) Assay

Following the protocol in *Tennessen et al., 2014*, we assayed total TAG levels normalized to total protein in whole male flies. To assay, flies were $CO_2$-anesthetized and flash frozen. Pairs of flies were homogenized in lysis buffer (140 mM NaCl, 50 mM Tris-HCl pH 7.4, 0.1% Triton-X) containing protease inhibitor (Thermo Scientific). Separation by centrifugation produced a supernatant containing total protein and TAGs. Protein reagent (Thermo Scientific Pierce BCA Protein assay) was added to the supernatant and the standards and incubated for 30 min at 37°C, then tested for absorbance at 562 nm on a Tecan Plate Reader Infinite 200. TAG reagent (Stanbio Triglycerides LiquiColor Test) was added to supernatant and standards, incubated for 5 min at 37°C, then tested for absorbance at 500 nm. n = 1 is two flies per homogenate.

## Proboscis Extension Response

Flies were fasted for 24 hr in a vial with a Kimwipe dampened with 2 mL of milliQ-filtered deionized (milliQ DI) water and tested for the proboscis extension response (PER) (*Shiraiwa and Carlson, 2007*). Water and all tastants were tested manually via a solution-soaked Kimwipe. Sucrose solutions were dissolved in milliQ water and presented in descending order by concentration. Groups of 10–15 flies were tested simultaneously. n = 1 equals a single animal.

## Imaging data analysis

For each fly, $\Delta F/F_0$ was calculated from a baseline of 10 frames recorded just prior to the stimulus (sucrose or water). Area under the curve (AUC) was calculated by summing the $\Delta F/F_0$ values from the initiation of the response to its end. Peak $\Delta F/F_0$ is the single maximum acquired within a response, and latency to peak was calculated by determining the time between the stimulus delivery and the peak response.

## optoFLIC Data Analysis

OptoFLIC analysis of daily food interactions, meal size, and meal duration was performed as previously described (*May et al., 2019*). R code used can be found on Github (https://github.com/chrismayumich/May_et_al_optoFLIC; copy archived at https://github.com/elifesciences-publications/May_et_al_optoFLIC; *May, 2020*). Briefly, food interactions were determined by calculating a

moving baseline on the raw data and selecting signals which surpassed threshold above baseline. These signals were then summed in 30 min bins. From the binned data, daily food interactions and the start and end of meals were calculated. The evening meal was used for all meal-based calculations to control for variability in meal shape. Meal size and duration were derived using meal start and end. Post-peak feeding duration was quantified as [(*time of meal end*) - (*time of meal peak)*].

An event is defined as a string of consecutive food interactions. R code used to extract event information can also be found on the Github link above. To calculate events per meal, the number of events between the meal start and meal end per meal were summed for each fly. Feeding rate was quantified as *[(events per meal) / (meal duration)]* per meal per fly. Pre- and post-peak feeding rates were quantified, using the time of the meal peak determined by food interactions, also used to calculate post-peak feeding duration, as *[(number of events pre- or post-peak) / (pre- or post-peak feeding duration)]*. Pre-peak feeding duration was quantified as *[(time of meal peak) - (time of meal start)]*.

## Split-*GAL4* expression overlays
Overlays created in Virtual Fly Brain (*Milyaev et al., 2012*).

## Acknowledgements

We thank Carrie Ferrario, Shelly Flagel, and Josie Clowney for helpful discussions and Scott Pletcher for his continuous assistance with the FLIC. We also thank all the researchers that shared protocols and fly lines (listed in the Materials and methods). Julia Kuhl designed some of the graphics for the manuscript. This work was funded by NIH R00 DK-97141 and NIH 1DP2DK-113750, a NARSAD Young Investigator Award, the Klingenstein-Simons Fellowship in the Neurosciences, and the Rita Allen Foundation (to MD), and by Training Grant T32-GM008322 (to CEM).

## Additional information

### Funding

| Funder | Grant reference number | Author |
| --- | --- | --- |
| National Institutes of Health | T32-GM00832 | Christina E. May |
| National Institutes of Health | R00 DK-97141 | Monica Dus |
| National Institutes of Health | 1DP2DK-113750 | Monica Dus |
| Brain and Behavior Research Foundation | NARSAD Young Investigator | Monica Dus |
| Esther A. and Joseph Klingenstein Fund | | Monica Dus |
| Rita Allen Foundation | | Monica Dus |

The funders had no role in study design, data collection and interpretation, or the decision to submit the work for publication.

### Author contributions

Christina E May, Conceptualization, Resources, Data curation, Software, Formal analysis, Funding acquisition, Validation, Investigation, Visualization, Methodology, Writing - original draft, Project administration, Writing - review and editing, Conducted all the experiments, with the exception of PER, Helped write the manuscript; Julia Rosander, Jennifer Gottfried, Evan Dennis, Investigation, Helped with the TAG measurements; Monica Dus, Conceptualization, Data curation, Formal analysis, Funding acquisition, Investigation, Visualization, Methodology, Writing - original draft, Writing - review and editing, Carried out the PER experiments and supervised the project, Wrote the manuscript

#### Author ORCIDs
Christina E May  https://orcid.org/0000-0001-9366-8592
Monica Dus  https://orcid.org/0000-0003-1465-9028

#### Decision letter and Author response
Decision letter https://doi.org/10.7554/eLife.54530.sa1
Author response https://doi.org/10.7554/eLife.54530.sa2

## Additional files

#### Supplementary files
• Transparent reporting form

#### Data availability
All data generated or analyzed during this study are included in the manuscript and supporting files.

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
