## [Decision Letter]

**Acceptance summary:**

In this manuscript, May et al. elaborate a central mechanism that potentially underlies increased sugar consumption in flies chronically fed high-sugar diets. They show that a high sugar diet, which decreases synaptic release from axon terminals of sugar responsive *Gr64f* neurons occurs together with decreased sucrose-induced activity in a PAM subclass of dopaminergic neurons that innervate the β'2 lobe of the Mushroom Body (MB). This reduction is DA neuron activity is shown in response to high sugar diet but not affected following high fat diet, arguing that the decrease in sugar evoked dopamine release accounts for a compensatory increase in consumption.

**Decision letter after peer review:**

Thank you for submitting your article "Dietary sugar inhibits satiation by decreasing the central processing of sweet taste" for consideration by *eLife*. Your article has been reviewed by three peer reviewers, including Mani Ramaswami as the Reviewing Editor and Reviewer #1, and the evaluation has been overseen by Catherine Dulac as the Senior Editor. The following individual involved in review of your submission has agreed to reveal their identity: Alex C Keene (Reviewer #3).

The reviewers have discussed the reviews with one another and the Reviewing Editor has drafted this decision to help you prepare a revised submission.

Summary:

Previous work by the authors May et al. (2019) showed that prolonged high-sugar diets resulted in increased food consumption together reduced sugar-evoked sensory responses in *Gr64f* neurons. Strikingly, increased feeding behaviour could be corrected by genetic or ontogenetic manipulations that increased *Gr64f* neuron response to sugar. In this manuscript, May et al. examine the central brain mechanisms underlying increased food consumption in flies chronically fed high-sugar diets. Specifically, the authors investigate how high sugar diet affects the function of Dopamine (DA) neurons that innervate the β'2 lobe of the Mushroom Body and the consequences of such functional changes in DA neuron activity. They show that a high sugar diet, which decreases synaptic release from axon terminals of Gr64f neurons in the SEZ, occurs together with decreased sucrose-induced activity in PAM-DA neurons that innervate the β'2 lobe of the Mushroom Body (MB). This reduction is DA neuron activity is shown in response to high sugar diet but not affected following high fat diet. Moreover, activating PAM-DA-β'2 neurons in flies that are kept on high sugar diet, rescues the diet induced obesity phenotype. The experiments are scientifically sound and the manuscript is well written. However, there are additional experiments authors need to do to support and clarify their findings and conclusions, particularly pertaining to the roles and mechanism of action of PAM-DA-β'2 neurons.

Essential revisions:

1) It is surprising that neuronal activation via some split-*GAL4s* that label β'2 lobes do not rescue the phenotype like MB301B. This needs to be rationalized. For instance the authors could perform additional experiments to clarify which β'2 neurons are labelled in MB301B line and show these neurons are not labelled in other splits. Such an anatomical characterization of different split-*GAL4s* innervating the β'2 should be made easier because all of these lines are registered to a standard brain atlas at Janelia, which should help one determine whether MB301B labels different DANs compared to the splits that do not rescue the phenotype. The expression pattern of MB301B on the Janelia Split Database suggests that this line also labels neurons outside of the Mushroom body; there are ascending or descending tracks seen in the ventral nerve cord. Could activation phenotypes seen in MB301 be due to these neurons that are not in the MB? The text should be revised to include a potential explanation for how dopamine release by different dopaminergic neurons onto the same target cells could have different acute effects on behavior.

2) Tsao et al. (2018) show that MB301 neuron activation leads to increased food search behavioural and their silencing in hungry flies results in reduced food search behaviour. This appears to contrast with findings here that the decreased activity in MB301 neurons (due to high sugar diet) causes an increase in food intake behavior over time. How do these two results fit together? This issue should be explicitly addressed. For instance it would be useful to know what impact of chronic activation of β'2 DANs is on feeding behaviour. Current activation protocol used in the experiments only activate the β'2 DANs when flies are eating. One would argue that chronic activation might lead to increased feeding behaviour potentially reconciling the two studies?

3) One point of concern is that the authors do not describe upstream or downstream neurons to the identified PAM-β' neurons. It is true that their upstream neurons are poorly defined, but it would be useful to know how responses of MBs or MB output neurons are impacted by feeding protocol and influence feeding behavior. The authors should therefore test if optogenetic activation or inhibition of MBONs that innervate the β'2. will rescue the high sugar diet phenotype as predicted by their analysis of PAM-β'2 DANs.

4) Several observations shown in in Figure 2 and 3 argue that the decrease in high sucrose-evoked DA neuron activity is directly linked to the altered responses in *Gr64f* neurons. This is nice to see but not unexpected as reduced *Gr64f* neurons output would be expected to decrease stimulation of downstream DA neurons. However the effect on *Gr64f* neuron synaptic release is far more drastic that the effect of high sugar diet on β'2 DA activity (50% suppression). How do the authors explain this difference?

5) In the first paragraph of the Results, the term "30% sugar" should be changed to "30% sucrose" if indeed sucrose is the only sugar that is used in these taste neuron stimulation experiments. (If not, the types of sugar should be indicated.) Is decreased synaptic release in *Gr64f* neurons observed in response to all sugars or only to sucrose?

6) What does "n number" indicate in these experiments? Is it the number of animals or the number of trials? In the control group, the fluorescence changes seem to have a bigger variation (the distribution looks like a bimodal distribution) then the high sugar diet experimental group. Is the variation coming from trial to trial differences or in between animal differences? I think this is an important point to clarify. Also, authors are using ∆F/F_0_ AUC values to compare the results of the experimental group and the control group. They don't explain why they choose to use this metric compared to peak ∆F/F_0_ value that is generally used to quantify fluorescent indicator responses. In the Materials and methods, it is also not clear how they choose the baseline F_0_. "For each fly, ∆F/F_0_ was calculated from a baseline of 10 samples recorded just prior to the stimulus". What does sample mean here? Do the authors mean data points? What is the speed of data collection? Please clarify and explain.

7) Several additional control or background experiments will be useful to better understand how the identified DANs regulate feeding behaviour. While all do not need to be addressed experimentally, at least the underlying ambiguities should be explicitly addressed in the text.

a) The conclusions would be strengthened with the inclusion of additional concentrations of sucrose. Does reduced responsiveness occur across concentrations or only to high concentrations of sugar? After being returned to normal diet for a number of days, does DAN activity return to normal?

b) Will PAM2 activation affect acute feeding behaviour in control flies at different concentrations (low-high) of sugar? (This seems important anyway.)

c) Could PAM2 activation affect acute feeding behaviour in control flies – potentially overriding low concentration of bitter compounds?

d) Will blocking the β'2 PAM neurons in control-diet fed flies result in increased feeding behaviour as predicted and concluded by the authors?

8) To really show that OGT works via its role in sensory neurons, the authors should ideally use targeted RNA interference to knockdown OGT in *Gr64f*+ neurons (which was used in their previous publication) and in the insulin producing cells, in order to make the point that changes in PAM-β'2 responses to sugar taste occur via changes in the sweet sensory neurons output rather than because of the metabolic side-effects of high nutrient density. Otherwise the conclusion should be duly qualified.

9) Please discuss the genetic background. It states that *W1118*;*Canton-S* was a control. Were lines outcrossed to this strain? Where did the control strain come from?

---

## [Author Response]

Essential revisions:1) It is surprising that neuronal activation via some split-GAL4s that label β'2 lobes do not rescue the phenotype like MB301B. This needs to be rationalized. For instance the authors could perform additional experiments to clarify which β'2 neurons are labelled in MB301B line and show these neurons are not labelled in other splits. Such an anatomical characterization of different split-GAL4s innervating the β'2 should be made easier because all of these lines are registered to a standard brain atlas at Janelia, which should help one determine whether MB301B labels different DANs compared to the splits that do not rescue the phenotype. The expression pattern of MB301B on the Janelia Split Database suggests that this line also labels neurons outside of the Mushroom body; there are ascending or descending tracks seen in the ventral nerve cord. Could activation phenotypes seen in MB301 be due to these neurons that are not in the MB? The text should be revised to include a potential explanation for how dopamine release by different dopaminergic neurons onto the same target cells could have different acute effects on behavior.

Thank you for this suggestion. We tested additional PAM DANs that innervated the same regions to see if the population labeled by *MB301B* was unique in rescuing the obesity phenotype. While *MB301B*, *MB109B*, and *MB032B* all innervate the β'2 lobe, they are considered distinct DANs – PAM #3, PAM #2, and PAM#6 respectively– that target different downstream cells (Aso et al., 2014). Thus, while innervation of the β'2 compartment is necessary, likely because that is where taste information arrives (Huetteroth et al., 2015; Yamagata et al., 2015), our data suggest that it is not sufficient, and that *MB301B* PAMs may have a special function. We agree that registering the lines and making this point clear is really critical for future studies, so we have used the Virtual Fly Brain (Milyaev et al., 2012) to do this. These data are now in Figure 4—figure supplement 2 and text was added to the Discussion to explain this point. Unfortunately, there is a glitch in the software and a comparison with *MB109B* (PAM #2) was not possible. We contacted Greg Jefferis about this and he said that this fix is complicated and would take a long time if at all possible, thus we have not included this particular line in the figure, but all the others are present, and this line is already considered distinct from *MB301B* (PAM#3) in Aso et al., 2014. We have found that *MB301B* does not have a perfect overlap with the other lines we tested and that it seems to have more ventral and anterior innervation in the mushroom body.

The reviewers are correct that we cannot exclude the possibility that VNC neurons have an effect on our phenotype. We have revised the text to address this.

2) Tsao et al., 2018 show that MB301 neuron activation leads to increased food search behavioural and their silencing in hungry flies results in reduced food search behaviour. This appears to contrast with findings here that the decreased activity in MB301 neurons (due to high sugar diet) causes an increase in food intake behavior over time. How do these two results fit together? This issue should be explicitly addressed. For instance it would be useful to know what impact of chronic activation of β'2 DANs is on feeding behaviour. Current activation protocol used in the experiments only activate the β'2 DANs when flies are eating. One would argue that chronic activation might lead to increased feeding behaviour potentially reconciling the two studies?

We apologize for the oversight in discussing Tsao et al. (2018) and Musso et al. (2019), we should have addressed their findings. Tsao et al. found that acute activation of *MB301B* neurons with *TrpA1* increased food seeking behavior, and that inactivation with *shibire^ts^* had the opposite effect. Musso et al. found that acute, closed loop optogenetic activation of the same line induced an increase in feeding interactions. Both of these studies were done in fasted flies and for a short time duration. In new experiments now described in Figure 4—figure supplement 1, we also found that closed loop activation of *MB301B* neurons in flies fed 5% sucrose on the FLIC for two days, led to higher feeding interactions. Thus, these neurons are clearly involved in feeding and food seeking behavior, and our data agrees with the overall findings of the two manuscripts when flies are on a control diet. However, we think that this is a distinct phenomenon than that described here, because in our manuscript we used optogenetic activation to *“*normalize*”* the decrease in sweet-taste responses in the presence of the stimulus, rather than to activate the neurons from their “resting” state in the absence of stimulation. The fact that we see a specific high sugar diet x *MB301B* effect suggests that our experiments are testing fundamentally different behaviors, even if the manipulations and the assays are similar (although our flies are not fasted and the FLIC runs for days, not minutes, and the animals are on a sugar diet). For example, in Musso et al., activation of *MB301B* induced more feeding interactions to both water and sugar, whereas in our case, the dietary environment in which the flies were on the FLIC, (5% vs. 20%) gave rise to different behaviors (increasing vs. stable feeding). Further, Musso et al. also tested MB056B: this line had an identical phenotype to MB301 in their assay, increasing the number of feeding interactions to both water and sugar on the FLYPAD, but the same line did not rescue diet-induced obesity in our experiments. Together these suggest that we are uncovering a circuit “state” by environment interaction, which is different from testing whether these neurons are sufficient for feeding. We have added this experiment and point to the Discussion. There we speculate that this state is connected to the expected level of reward (from sweetness), and that stable processing of the taste signal centrally is essential to stabilize eating. Thus the relative changes in PAM signaling, rather than their absolute value, is important. This seems to fit well with current interpretations of DAN function. In this scenario, when we stimulate the *MB301B* DANs in flies on a high sugar diet we are addressing the change that occurs with diet and keeping their activity consistent.

3) One point of concern is that the authors do not describe upstream or downstream neurons to the identified PAM-B' neurons. It is true that their upstream neurons are poorly defined, but it would be useful to know how responses of MBs or MB output neurons are impacted by feeding protocol and influence feeding behavior. The authors should therefore test if optogenetic activation or inhibition of MBONs that innervate the β'2. will rescue the high sugar diet phenotype as predicted by their analysis of PAM-β'2 DANs.

We agree with the reviewer/s that this is an incredibly interesting question, but we disagree that it should be answered in the current manuscript. First, we feel that this question deserves its own in-depth study, where the impact of DANs alterations on MBON plasticity, learning and memory, feeding and obesity are investigated. Indeed, this is the thesis project of another PhD student in the lab, - who has been setting up these assays and tackling different parts of this question over the last 2 years. 4) Several observations shown in in Figure 2 and 3 argue that the decrease in high sucrose-evoked DA neuron activity is directly linked to the altered responses in Gr64f neurons. This is nice to see but not unexpected as reduced Gr64f neurons output would be expected to decrease stimulation of downstream DA neurons. However the effect on Gr64f neuron synaptic release is far more drastic that the effect of high sugar diet on β'2 DA activity (50% suppression). How do the authors explain this difference?

We believe this is due to the differences in the dynamic ranges of the fluorescent probes. GCaMP6 more faithfully reports even small fluctuations of intracellular calcium, while syb-phluorin is unable to capture smaller fluctuations in signal from vesicular release. For example, we measured a 30% decrease in the Gr64f+ neural response to 30% sucrose stimulation of the proboscis in May et al., 2019, which is consistent with the GCaMP6 fluorescence changes measured in the *MB301B* neurons here. However, it is also possible that some of the “decrease” in the sweet signal is compensated for along the way. Unfortunately, without a genetically defined anatomy of the taste processing circuit between the *Gr64f*+ and PAM neurons, we cannot answer this question.

5) In the first paragraph of the Results, the term "30% sugar" should be changed to "30% sucrose" if indeed sucrose is the only sugar that is used in these taste neuron stimulation experiments. (If not, the types of sugar should be indicated). Is decreased synaptic release in Gr64f neurons observed in response to all sugars or only to sucrose?

Thank you for pointing this out, we changed it to 30% sucrose. We never published the data, but found that a high sucrose diet led to a decrease in taste responses to D-glucose and L-glucose.

6) What does "n number" indicate in these experiments? Is it the number of animals or the number of trials? In the control group, the fluorescence changes seem to have a bigger variation (the distribution looks like a bimodal distribution) then the high sugar diet experimental group. Is the variation coming from trial to trial differences or in between animal differences? I think this is an important point to clarify. Also, authors are using ∆F/F_0_ AUC values to compare the results of the experimental group and the control group. They don't explain why they choose to use this metric compared to peak ∆F/F_0_ value that is generally used to quantify fluorescent indicator responses. In the Materials and methods, it is also not clear how they choose the baseline F_0_. "For each fly, ∆F/F_0_ was calculated from a baseline of 10 samples recorded just prior to the stimulus". What does sample mean here? Do the authors mean data points? What is the speed of data collection? Please clarify and explain.

We have answered the questions below and further clarified it in the Materials and methods.

For calcium imaging experiments, n counts each ROI, of which there are 2 per fly

For the FLIC, n counts individual flies

For metabolic assays, n counts single homogenates of two flies each

The variation in calcium imaging comes from between animal differences. When we compared the Peak % ∆F/F_0_ responses to sucrose on different days of imaging we found no differences (One-way ANOVA with Kruskal-Wallis multiple comparisons). Although each animal was placed on the diet at the same time and treated identically, there are still variations in how much they eat relatively to one another per day, the extent of their taste changes, and also in their metabolism (May et al., 2019).

We chose to calculate ∆F/F_0_ AUC instead of peak ∆F/F_0_ for the synaptophluorin imaging because looking at individual traces it was often difficult to unambiguously define a peak. Thus we felt that ∆F/F_0_ AUC would be a more direct way to show these data.

We changed “sample”, to “frame”, one frame in our experiments was captured at 4Hz as listed under methods.

7) Several additional control or background experiments will be useful to better understand how the identified DANs regulate feeding behaviour. While all do not need to be addressed experimentally, at least the underlying ambiguities should be explicitly addressed in the text.a) The conclusions would be strengthened with the inclusion of additional concentrations of sucrose. Does reduced responsiveness occur across concentrations or only to high concentrations of sugar? After being returned to normal diet for a number of days, does DAN activity return to normal?

We have carried out the calcium imaging experiments with 5% sucrose stimulation to the proboscis. We see a similar decrease in calcium responses to this lower concentration when flies are fed a high sugar diet. This is consistent with the findings that PER is decreased to both low (1% and 5%), mid (10%), and high (20% and 30%) sucrose (May et al., 2019). These data are now included in Figure 3—figure supplement 1C and D.

We were unable to carry out recovery experiments, but have a manuscript on bioRxiv showing that taste deficits last even after flies were moved to a control diet for 10 days

(doi.org/10.1101/2020.03.25.007773). As such, we hypothesize that PAM responses should also remain low. We plan to follow up independently on these findings and look at the effects of long term changes in taste and reward signaling on feeding behavior and obesity.

b) Will PAM2 activation affect acute feeding behaviour in control flies at different concentrations (low-high) of sugar? (This seems important anyway.)

We have conducted OPTOFLIC experiments with 5% food for two days (see point #2). We find that acute activation of flies on a control diet and fed 5% sucrose food (similar to the control diet concentration) on the FLIC results in an increase in feeding behavior, as others have reported (Musso et al., 2019 showed increased feeding with both water and sugar). We have added these data to Figure 4—figure supplement 1 and this point to the Discussion.

c) Could PAM2 activation affect acute feeding behaviour in control flies – potentially overriding low concentration of bitter compounds?

*MB301B* activation does increase feeding behavior acutely (see response to point #2), but we have not tested whether it can overcome bitter compounds; while this is an interesting question, we feel that given the current circumstances, this is not directly related to the central hypothesis of the study. We have added this point to the Discussion, however.

d) Will blocking the β'2 PAM neurons in control-diet fed flies result in increased feeding behaviour as predicted and concluded by the authors?

We have carried out preliminary optogenetic inhibition experiments using *UAS-GtACR*. Using a moderately sweet food, 8% sucrose, we have observed an increase in feeding behavior in *MB301B > UAS-GtACR* flies but were unable to complete more experiments due to COVID-19. There is a long history and debate on the role of DANs inhibition in feeding behavior. Expectations from D2R genetic studies and the reward deficit theory would argue that deficits in DAN signaling should result in increased eating, but many observations show that inhibition of DANs via pharmacology decreases feeding instead. This is likely because the levels of inhibition and the context in which this is delivered (the dietary history and experience of the animal) matter. Given that we were not able to address these points to the level of depth they deserve, and that GtACR doesn’t have the consistency in performance that CsReacher or Chrimson have, we would prefer not to add them to the main manuscript at this time.

8) To really show that OGT works via its role in sensory neurons, the authors should ideally use targeted RNA interference to knockdown OGT in Gr64f+ neurons (which was used in their previous publication) and in the insulin producing cells, in order to make the point that changes in PAM-β'2 responses to sugar taste occur via changes in the sweet sensory neurons output rather than because of the metabolic side-effects of high nutrient density. Otherwise the conclusion should be duly qualified.We agree with the reviewers, and this was the experiment we originally planned, but we have been unable to build a stock with the 5 transgenes in part because of impossibility of recombining some of them, since they were integrated in the same attp site (2 copies each of the split GAL4s to ensure expression of both UAS, UAS-GCaMP,UAS-OGT RNAi). We have modified the text to point out this caveat in the text and in the Discussion. We have found that knock down of OGT has no effect on fat accumulation in *MB301B* neurons alone (Figure 3—figure supplement 2).

9) Please discuss the genetic background. It states that W1118;Canton-S was a control. Were lines outcrossed to this strain? Where did the control strain come from?

The *w1118-Cs* strain was made from Anne Simon in the Benzer lab by backcrossing *w1118* to *Canton S* for 10 generations. This information is now in the Materials and methods.